# Additive Manufacturing and Mechanical Characterization of PLA-Based Skull Surrogates

**DOI:** 10.3390/polym15010058

**Published:** 2022-12-23

**Authors:** Ramiro Mantecón, Miguel Marco, Ana Muñoz-Sanchez, George Youssef, José Díaz-Álvarez, Henar Miguélez

**Affiliations:** 1Department of Mechanical Engineering, Universidad Carlos III de Madrid, 28911 Leganés, Spain; 2Experimental Mechanics Laboratory, San Diego State University, San Diego, CA 92182, USA

**Keywords:** head impact, surrogate, polymer 3D-printing, PLA

## Abstract

Several occupational and leisure activities involve a high risk of head impacts, resulting in varying degrees of injuries with chronic consequences that adversely affect life quality. The design and manufacturing of effective head protections rely on proper head simulators to mimic the behavior to impact loading. 3D-printed human skulls are reported herein to address the need for reproducible, cost-effective, anatomically-correct surrogates. To demonstrate the viability of the investigated approach, surrogate bone sections and skulls were mechanically tested under quasi-static loading conditions. The 3D-printed bone sections were flexural tested, elucidating the effect of printing orientations and the sample geometry on their mechanical behavior. The printing orientation minimally influenced the results due to the high infill percentage, while the sample geometry played a major role in the flexural properties because of the change in the section properties. The surrogate skulls were submitted to lateral compression and frontal penetration tests to assess the impact of the sectioning strategy on the overall mechanical performance. Results indicate that PLA-based surrogates reasonably reproduce the behavior of skulls. In addition, the sectioning strategy elucidated the effect of skull sutures, while streamlining the additive manufacturing process. The outcomes lay the foundation for future research seeking a complete surrogate head.

## 1. Introduction and Motivation

The health hazards from brain injuries are a primary societal concern, resulting in significant health consequences that can affect the quality of life for patients and their extended families. Brain injuries are commonly associated with head trauma due to blunt impact during violent accidents or biomechanical loading scenarios from occupational or leisure activities [1]. The concerns derived from head trauma arise from severe, dynamic load impositions, resulting in severe jolting of the brain matter, in what is referred to as concussion or Traumatic Brain Injury (TBI). TBI has serious repercussions regardless of the injury source, whether in military or civilian contexts. For example, traumatic injuries are the highest fatality caused in industrialized countries for ages under 45 [2]. However, in vivo optimal impact mitigation mechanisms are under-investigated because of the obvious ethical concerns. Therefore, the development of head surrogates is imperative for revealing the underlying mechanistic deformation mechanisms and accelerating innovations to effectively mitigate the severity of head trauma, e.g., using head protection such as helmets.

While most symptoms of mild Traumatic Brain Injury (mTBI) disappear in the first few weeks after the incident, short- and long-term consequences, including chronic symptoms, are a growing concern [1,3]. This concern required a call to action, promoting scientific, technological, and industrial reexamining of current impact mitigation mechanisms. Helmets are the primary structures used to mitigate the adverse effects of head trauma, and they are widely used in all the sectors mentioned above [4,5,6,7]. However, the design and development of effective protections require a profound knowledge of the injury mechanisms while observing, analyzing, and evaluating the protective structures in congruent operating conditions. This call for action is further challenged by the infeasibility of in vivo testing, requiring the development of head surrogates with similar mechanical behavior to their biological counterparts. Mechanical comparability resides in several metrics, such as elasticity, impact resistance, and energy absorption capacity, to name a few key performance metrics [8]. The latter are used to develop instrumented or passive head dummies, which are commonly utilized as anthropometric simulators in laboratory testing of different biomechanical impact scenarios. In other words, these anthropometric head surrogates eclipse the need for in vivo testing and accelerate the development of protective structures with higher impact efficacy. This necessitates obtaining more accurate experimental results, the validity of which hinges on the viability of the testing surrogates and methods used in simulating head trauma. Here, the practicality of head surrogates centers around geometry replication, comparable material properties, and rapid manufacturing. The geometry plays an important role in impact mechanics, deciding the load–structure interactions such as load path and distribution based on the localized anatomical features with respect to the load application site. Therefore, anatomical correctness is essential to faithfully reproduce the adverse effects of head trauma. The levels of induced stresses and strains during loading hinge on the dynamic nature of the loading scenario and the time-dependent behavior of biological materials [9,10]. Finally, increased testing repetition requires accessible and rapid manufacturing techniques to produce head surrogates quickly. Recent advances in additive manufacturing methodologies are an appealing area for developing head surrogacy using polymers [11,12].

The preceding discussion stipulated the practical requirements of head surrogates based on foundational mechanics considerations, e.g., geometry, material, and manufacturing, commonly referred to as process–structure–property interrelationships [13]. However, accounting for failure mechanisms is imperative given their severe outcomes. Head trauma is linked to a variety of failure mechanisms, from skull fracture to brain injury [14,15]. From a mechanistic standpoint, several metrics are used to assess the potential and extent of failure. Linear and rotational accelerations result in compression and violent rotation of the brain matter. The latter can be the source of internal hemorrhage due to tearing of the meninges, the connective tissues between the cranium and brain matter. Arguments exist concerning the contribution of angular accelerations and their linear counterparts to TBI during impact events [16,17]. On the other hand, the intracranial pressure and associated deformations are primarily responsible for skull fractures. Skull fracture is one of the most severe and common consequences of impacts within the wide range of head injuries. Therefore, developing effective head protection mechanisms depends on the accuracy of realistic physical models of the human head to facilitate the probing and detection of the failure limits using partial or complete surrogates. Hence, the mechanical properties and geometrical description of the surrogate are essential performance metrics to assess the damage indicators of TBI-associated failures [3]. The abovementioned considerations serve as the motivation for the research leading to this report, seeking to the overarching objective of this project to accelerate the realization of skull surrogates using advanced manufacturing techniques.

The pursuit of representative head elements, including the scalp, cranium, and brain matter, has been previously reported in the open literature [18,19,20]. In such a pursuit, the geometrical descriptions are the most challenging, followed by material choices. Several researchers opted for minimum, resembling geometries, while others sought an accurate description relying on advances in reverse engineering technologies, as is the case of the current research. Using an advanced engineering approach is associated with a notable increasing in effort and cost, but it recently became feasible with the current paradigm shift in manufacturing complex geometries using additive approaches. In the past, manufacturing approaches hindered steady progress in developing real-life simulants; hence, researchers focused on creating a minimal anatomic resemblance to the human head to accommodate the limited capabilities of subtractive manufacturing in realizing complex geometries. For example, Ward et al. [21] proposed a reduced order solution of Nahum et al. [22] to validate the experimental results of measurements of intracranial pressure. More recently, other authors used spherical or ellipsoidal-shaped surrogates to overcome the challenges of fabricating and testing head surrogates [23,24]. They also utilized glass–epoxy mixtures for the skull construction and silicone gels for the brain. Moving beyond simplified representation, Taha et al. [25] presented an anatomically similar plastic-based head surrogate. However, the model biofidelity was compromised because the gel-filled skull constrained brain motion. Polymers (hydrocarbons and silicone) are commonly used given their structural and mechanical resemblance to biological materials, which is further facilitated by polymer additive manufacturing. While several publications exist that address quasi-static loading in flexural or tensile specimens [26,27,28], there is a gap in understanding the mechanical behavior of head surrogates under quasi-static loading conditions (the underlying motivation of the current study).

The overarching objective of this research is to demonstrate the use of 3D printing techniques to develop a human skull surrogate, taking advantage of the resources provided by these technologies in accessibility, customization, and repeatability. Surrogates were printed using poly (lactic acid) (PLA) since it is the most commonly used in the 3D printing community. PLA is also readily accessible, easily printable, and affordable, and available publications highlighted additional promising use of PLA for bone-resemblance applications [29,30,31].

## 2. Materials and Methods

### 2.1. Polymer 3D Printing

The fused filament fabrication (FFF) technique is based on the extrusion of a molten polymer filament [32]. Material deposition follows a layer-by-layer scheme, where the previously deposited material is regarded as the substrate for the freshly laid material. This 3D printing process is thermally driven; therefore, the temperature of the different components, including the printing nozzle, printing bed, and printing environment, play a major role in the outcome. In general, the material quickly solidifies and retains its shape. The temperature differences between the already cool substrate material and subsequent layers establish thermal gradients that are paramount to layer adhesion and, finally, to the mechanical properties of the printed structure [33,34]. All samples used herein were fabricated using Epsilon W50 3D Printer by BCN3D. The printing temperature was set to 210 °C, and building surface temperature was set to 60 °C, i.e., standard printing conditions for PLA. Layer height was set at 0.3 mm and all other printing parameters were left at default values for PLA, including a printing speed of 60 mm/s.

### 2.2. Printing Material (PLA)

PLA is the regarded as the most common material in FFF 3D-printing due to its low printing requirements, including open fabrication enclosure [35]. It has been noted as a viable petrochemical replacement polymer, with desirable biodegradability properties [36]. Futhermore, PLA has earned the classification of “Generally Recognized as Safe” from the United States Federal Drug Administration, leading to several biomedical applications [37,38]. Moreover, the mechanical properties of PLA are generally competitive with conventional polymers with the unique advantage of tailoring the surface properties by altering the reactive groups to change hydrophilicity or surface chemistry [39,40]. In addition, PLA has been vigorously investigated in additive manufacturing applications, including the interrelation between the quality of the final part and the printing parameters. For example, Song et al. [41] extracted specimens from a fully dense block and noted the superior mechanical performance of PLA compared to other thermoplastics. Other researchers focused on the effect of layer height and printing speeds on the quality and performance of the final printed parts [42,43,44]. In combination with other materials and biological agents, PLA was also used to manufacture scaffolds for regenerative medicine [31,38,45,46]. In all, PLA is a 3D printable, biocompatible, and property-tailorable polymer that also found utility in biomedical applications such as bone substitutes [29,47]. Therefore, ivory while PLA (Smart Materials 3D, SMPLA0WH0B075) was used in the current study to fabricate skull surrogates, as discussed next.

### 2.3. Skull Bone Flexural Specimens

Flexural behavior of materials has been proved relevant concerning head impacts and damage in the literature, e.g., Delye et al. [48]. The flexural properties of materials depend on the geometrical attributes of the tested samples, highlighting the importance of replicating skull bone geometry in laboratory settings. It follows that the flexural properties of the surrogate skull bones, i.e., 3D printed samples, also hinge on the geometry based on the location of the section extraction as well as the age of the skull (subject age). Of equal importance is the 3D printing orientations. Merging the study by Lee et al. [49] and these considerations resulted in a test matrix inclusive of age, skull location, and printing orientations. The sections were extracted from different locations from their respective skulls [49], testing the flexural properties of bone sections from the skulls of two subjects at 61 and 86 yr. Specifically, Lee et al. [49] reported the geometries of bone sections from physical skulls, of which the thickness and depth are listed in Table 1, the thickness of skull bone being in the common range for 3-point bending tests, as indicated by Eisová et al. [50]. Therefore, two sample types representing the bare bone (referred to as ‘bare’) and soft tissue-covered bone (denoted as ‘covered’) were printed and characterized. Since the print orientation effect on the resulting flexural properties was emphasized, one set of samples was printed on one of its flat surfaces while the other was printed on one of the curved surfaces. The sample configurations are hereafter denoted as ‘flat’ and ‘vertical’, respectively. In all, 3D printed surrogate bone sections were from different skulls (skull 1 of a 61 yr subject, and skull 2 of an 86 yr subject), segregated locations (bare vs. covered), and print orientations (flat and vertical), resulting in a total of eight different sample sets. Each set consisted of three specimens or a total of 24 flexural specimens. All 24 samples were printed on an Epsilon W50 3D Printer using PLA with 100% infill. Irrespective of the sample configuration and type, the nominal span was set to 40 mm, and the nominal depth was set to 10 mm. Table 1 summarizes the flexural dimensions based on the printed samples.

### 2.4. Skull Manufacturing

The overarching objective of this study was to fabricate and characterize a 3D-printed, full skull surrogate. A companion goal was to allow the insertion of the brain matter surrogate, providing a pathway to realize a complete head replica for future studies. Therefore, splitting the skull based on one of the anatomical planes was preferred over sectioning or drilling after printing to ensure structural integrity. The three possible digital cuts that split the head into halves (sagittal, coronal, or axial) were considered. The sagittal cut yielded two symmetrical hemi-skulls. A coronal cut divided the skull in front and back halves, while an axial cut resulted in superior and inferior halves. The axial sectioning strategy was discarded because of the lack of a suitable testing approach for assessing the mechanical and structural performance of the assembled skull in biomechanical loading scenarios. Additionally, one un-sectioned skull was 3D printed using the same printing settings and was used as control to ascertain the effect of printing strategy on the mechanical and structural performance of the skull surrogates. Figure 1 shows the three final design strategies used in this research. Skull geometry was obtained from a realistic model available in an online repository (grabcad.com accessed on 13 September 2021). The internal structure (bone-diploë-bone) was simplified to a single material structure. Two skulls were additively manufactured using FFF approach for each strategy. Skull was printed as a hollow shell with a 4 mm wall thickness. No inner support was used in any case since the curvature of the skull allowed for stable adhesion of the layers. For the sagittal-cut, support was not needed in any other location of the structure. Coronal-cut required printing structures to support the maxilla. The whole skull required outer supports for the orbital socket. Adhesion of the skulls was done with cyanoacrylate epoxy due to its chemical and mechanical compatibility with PLA.

### 2.5. Mechanical Testing

For the mechanical testing, all samples (partial sections and full skulls) were tested on an Instron Universal Testing Machine 3366 equipped with a 10 kN load cell, i.e., the maximum load capacity. The partial section samples underwent flexural 3-point bending testing configuration. The fully assembled skulls were tested in compression and frontal point-loading.

#### 2.5.1. Three-Point Bending Tests

The 3-point bending tests were carried out at 10 mm/min, emulating the procedure outlined by Lee et al. [49] in quasi-static conditions and supressing any time- and rate-dependent deformation mechanisms [51]. A specific rig was designed, manufactured, and coupled to the testing machine for these tests. Figure 2 shows the testing setup, where a curved sample is loaded onto the testing fixture. Support beams capped with 1 mm diameter rods were located 30 mm apart, setting the testing span and leaving a 5 mm margin on each side. The loading arm was terminated with a 2 mm diameter rod and applied at mid-span. A preload of 1 N was set to eliminate slack and avoid testing variability. Following Lee et al. [49], the bending stress (σ) and the corresponding strain (ϵ) were calculated using the simplification of a prismatic beam, based on the applied force and the resulting deflection using Equations (Equation 1) and (Equation 2), respectively,
(1)$$\bar{\sigma }=\frac{3}{2}\frac{\bar{F}\times L}{b\times d^{2}},$$
(2)$$\bar{\epsilon }=\frac{6\times \bar{D}\times d}{L^{2}},$$
where *F* represents the applied force, *L* is the span set to 30 mm, *b* is beam depth as tabulated in Table 1, and *d* represents the thickness (also listed in Table 1). Flexural moduli and strengths were obtained from the calculated stresses and strains and compared to the reported results [49]. The flexural moduli were evaluated in the linear region of the bending stress-strain curves at strains between 2.25–3.25%.

#### 2.5.2. Skull Compression and Point-Loading Testing

Two testing procedures were devised to assess the mechanical response of the 3D-printed skulls, namely lateral compression and frontal point-loading. These tests were used to analyze the behavior of the final component under quasi-static loading.

A comparison of the effect of the manufacturing strategies was also performed. Sagittal-cut skulls were employed for lateral compressive tests, while a frontal point-loading test was performed on coronal-cut skulls. A 3D-printed whole skull was also tested using each procedure, and the results were compared. A total of three skulls (two cut and one whole) were subjected to each test. The lateral compressive test consisted of placing the printed skull on its side while applying the compressive force until failure. A standard set of compression platens with 190 mm diameter, allowing maximum contact with the skull throughout the test, was used for these tests, as shown in Figure 3A. On the other hand, Figure 3B shows the frontal point-loading testing setup, where the skull (assembled or whole) was positioned facing upward. The skull was laid on the lambdoid suture while the load was applied to the frontal bone. The penetrating tool was designed to have a tip diameter of 20 mm and a total length of 30 mm.

## 3. Testing Results and Discussion

The following three subsections correspond to the three different testing procedures discussed above. Results are categorized into 3-point bending tests for the flexural specimens, lateral compressive tests for the sagittal-cut skulls, and frontal point-loading tests for the coronal-cut skulls.

### 3.1. Flexural Behavior

Figure 4A includes the force-displacement plots from flexural testing of flat, 3D-printed bone surrogate samples with four different thicknesses, representing bone sections extracted from skulls from two age groups. Figure 4B reports the flexural results of vertically 3D-printed samples. For each sample, the average force-displacement response is bounded by a shaded area corresponding to the standard deviation, at a confidence interval of 95.5%, based on three separate measurements. In general, the flexural results in Figure 4 are mechanistically consistent, where samples with larger thicknesses are superior to their reduced thickness counterparts, irrespective of the printing orientation. For example, the average force-displacement curve of a flat sample with 8.40 mm thickness exhibits fourfold increase in the force at maximum displacement compared to another flat sample with 5.75 mm thickness. The same difference in the thickness resulted in a twofold increase in the force at maximum displacement in vertically printed samples, highlighting the first effect of the printing orientation on the mechanical behavior. For flat printed samples, a slight difference in the thickness yielded a significant change in the force-displacement response, where a reduction from 8.40 mm to 7.32 mm, i.e., ca. 1 mm, resulted in a 17.2% drop in the force at maximum displacement. On the other hand, the drop in the force at the end of the test was proportional to the thickness reduction, as shown in Figure 4B. In all, Figure 4 manifests the effect of print orientation on the overall behavior and the printing-induced sample-to-sample variations. The effect of printing orientation was minimal on average, as shown in Figure 5. This figure provides a qualitative comparison between the flexural responses as a function of the printing orientation.

The flexural force-displacement responses, shown in Figure 4, were converted to stress-strain curves using Equations (Equation 1) and (Equation 2) to calculate the flexural moduli and strengths as a function of sample geometry and print orientation. Table 2 summarizes the flexural properties, including modulus and strength for each sample type. Also reported in the same table are the respective properties from testing of physical skull bone samples based on the work of Lee et al. [49], as well as the relative difference of the former and their counterparts reported herein. The flexural moduli of surrogate bone samples were found to be independent of the printing orientation and thickness, except the modulus of the 5.75 mm samples. The same independence is also reported for the flexural modulus with similar exceptions. The flexural strength of the thin samples (i.e., 5.75 mm) was 102 MPa and 133 MPa for flat and vertically printed samples, respectively, indicating a 30% difference based on the print orientation. However, all remaining samples exhibited a significantly smaller difference in the strength values when comparing flat and vertically printed results. The departure of the flexural behavior of the thin samples is attributed to amplified printing-induced anisotropy by 20–30% reduction in thickness compared to other tested samples. The printing-induced anisotropy has also been reported for other materials [52].

At this point, the flexural properties of skull bone surrogates are compared to their natural counterparts previously reported in [49]. While the flexural properties of PLA-based samples varied significantly from those measured by Lee et al. [49], two noteworthy observations exist. First, the properties appear within the same order of magnitude as the results from flexural testing skull bone sections, confirming the suitability of PLA as a mechanical simulant of human bones [29]. Notably, the flexural modulus of 7.83 mm surrogate samples was in excellent agreement with their natural counterparts, reporting merely a ∼2% difference, irrespective of the printing orientation. The relative difference between synthetic and natural moduli exceeds 60% for samples with higher thicknesses, implying that the microstructure of the bone plays a major role in the flexural properties. In a similar study, Ondruschka et al. [53] reported the flexural properties of epoxy resin-based samples, stating a similar difference to those reported herein.

In the current study, the cross-section of the printed samples was homogenized, neglecting the composite structure of natural skull bone consisting of a layer of cancellous (spongy) bone sandwiched between two layers of compact (stiffer) bone. From a mechanics standpoint, the difference between the homogenized and sandwich structure maps to the variability between printed surrogate and natural bone behavior in thicker samples given the change in the distance from the neutral axis of the samples and their associated eccentricity [54]. Future studies should focus on replicating the microstructure of natural foams while exploring an alternative 3D printing approach, e.g., resin-based printing. Second is the relative inter-variability of the flexural moduli of PLA-based surrogate bone compared to those of their natural counterparts [49]. As stated above, the printing-based variability was controlled through sample homogenization for most of the samples investigated in this study (refer to the summary of results in Table 2). At the same time, the properties of natural bone exhibited a broader range of variability. The latter is attributed to (1) the inhomogeneous cross-section of skull bone, as discussed above, and (2) the inherent dichotomy in biological samples that have been optimized to sustain incoming loading [55,56,57].

### 3.2. Lateral Compression Results

Two sagittal-cut skulls and one whole skull, all 3D-printed using PLA, were tested in lateral compression, as shown in Figure 3A, where the compression platen was driven into the skulls until failure. Figure 6 shows the compressive force vs. displacement results for all three skulls. The sagittally sectioned skulls reported similar force-displacement plots, with a slight shift in the displacement at breakage. In general, the compressive force-displacement response of the sectioned skulls can be divided into two regions, namely structural and material elasticity regions, separated by a rapid strain hardening effect. The initial structural elasticity region extended from the onset of loading to the ∼9 mm for skull 1 and ∼11.5 mm for skull 2. At this point, the strain hardening momentarily took over the response by rapidly increasing the force at a moderate rise in compressive displacement. It is worth noting that the structural response region showed signs of sequential cracking, manifested as sudden drops in the force. The jagged behavior superposed on the force-displacement curves in the structural region is attributed to the inter-layer bonding failure of the 3D-printed structure. This is because of the minimal contributions of the material to the overall response up to this point. In the final region, i.e., the material elasticity region, the compressive force-displacement response continued to monotonically increase till ultimate fracture without noticeable signs of cracks as was the case in the structural region.

On the other hand, the compressive force-displacement of the whole 3D-printed skull was drastically different from their sectioned companions, terminating at ∼2500 N and 6.3 mm, as shown in Figure 6. The compressive behavior of the whole skull can be generally characterized as linear, with a single sudden drop in the force at ∼4.5 mm, corresponding to the initiation of the damage. Once the initiated crack was arrested, the compressive force-displacement response continued in a similar linear fashion. The mechanical behavior of the whole skull obscured the polymer contributions while giving rise to the structural effects manifested from the complex skull geometry. This is mechanistically intriguing, but not surprising, where the sectioned skulls outperformed the whole skull since the latter suffered from improved structural stiffness due to the absence of adhesion lines. The sectioned skulls are anatomically similar to their natural counterparts since the structural stiffness of the latter benefits from the sutures connecting the bony structures of the skull. In essence, the sectioning strategy adopted herein did not only improve the manufacturability of these head surrogates, but also, and arguably more important, implicitly elucidated the effect of sutures on the overall mechanical behavior of the skulls while demonstrating a path forward to additively manufacturing realistic head surrogates.

The stiffness was obtained from the slope of the force-displacement curves, shown in Figure 6, for all three skulls, listed in Table 3, reporting similar values of 491.1±9.6 N/mm. However, the maximum force differed based on the sectioning strategy, where the sagittal cut skulls reported a maximum force of ∼8000 N, whereas the whole skull suffered catastrophic failure before reaching 2500 N.

None of the 3D-printed skulls initially fractured along the adhesion line. The fracture initiated elsewhere between layers. However, once the crack front reached the adhesion zone, it expectedly followed the separation line. The incremental progression-arrest of the crack is captured in the final broken skulls, as shown in Figure 7A, where the crack front changed direction multiple times during testing (illustrated as a wavy black line in Figure 7A). These fracture patterns are consistent with the results obtained by [48] for low velocity, albeit different in the testing strategy. The difference in the fracture behavior between adhered and whole skulls is a consequence of process-induced anisotropy. The compressive force was coaxial with the surface vectors of the layers in the sagittal-cut specimens due to printing orientations. On the contrary, the compressive force was parallel to the layers in the whole skull, triggering an earlier breakage due to the shear stresses in the interface between layers. In other words, the printing orientation played a major role in the mechanical behavior of the skulls by altering the anisotropy, which has been a focus of other researchers through post-processing steps or shifting to other 3D printing processes [58]. Nonetheless, mechanical anisotropy is an inherent attribute of biological materials. Hence, biomimetic structures attempting to imitate the behavior of bone tissue should also exhibit comparable anisotropy [59,60].

### 3.3. Frontal Penetration Results

The coronal sectioning skulls presented similar force-displacement plots (Figure 8), starting with a quasi-linear region that transitioned into a plastic deformation region. The quasi-linear behavior at the onset of loading was also observed in the whole skull results. However, as the mechanical response transitioned into the plastic region, the response of coronally sectioned and whole skulls differed significantly. The latter continued along its initial hardening response, quasi-monotonic force-displacement behavior, until the first sign of plastic deformation (yield). The yield deformation is captured in the mechanical response as a precipitate cliff from sudden plasticization, i.e., momentary strain softening behavior [51,61], happening around 12 mm for the full 3D-printed skull. Thereafter, the force-displacement continued in a similar hardening behavior before reaching a maximum force at 19.21 mm. Shortly after, the full skull test was terminated due to testing limitations, but the skull remained intact, as shown in Figure 9B. On the other hand, the coronal cut skulls that were epoxy-adhered after 3D printing two halves exhibited a drastically distinct plastic region, generally characterized by underlying softening behavior. In the plastic region, the epoxy-adhered skulls generated a jagged force-displacement response due to the crackling of the skull. Like the whole skull results, the epoxy-adhered skull also displayed a sudden cliff due to the initiation of a circumferential crack at large displacements corresponding to layer debonding, ranging between 14–14.5 mm. The similarities in the mechanical response between the coronal cut skulls point to the repeatability and consistency of the additive manufacturing process. All specimens exhibited a permanent depression in the load application zone, as shown in Figure 9.

Table 4 lists the stiffness, maximum load, and displacement at yielding of the coronally sectioned skulls compared to the same attributes for the whole skull. Irrespective of the sample configuration (sectioned vs. whole), the average stiffness was 248.4±26 N/mm. The whole skull sustained the highest load at 1756 N, representing 530 N, on average, above those recorded for the coronally sectioned skulls.

The difference in the maximum attained force is attributed to the orientation of the adhesion lines to the load application site and the printing orientation. The coronally sectioned skulls were printed in the posterior–anterior direction for the facial portion of the skull, while the remaining section was printed in the anterior–posterior direction. This indicates that the bonding lines between layers were orthogonal to the loading direction. On the other hand, the full skull was printed in the inferior–superior direction, resulting in printing layers at an oblique angle with the loading direction (i.e., combined normal and shear forces). In all, it is important to note the differences in the failure between the sagitally cut skulls (Figure 7A), coronally cut skulls (Figure 9A), and whole skulls (Figure 7B and Figure 9B). Such differences signify the interrelationship between the printing parameters, the mode of loading, the skull geometry, and the material properties.

## 4. Conclusions

This research demonstrated an FFF-based printing approach to manufacture surrogate human skulls for laboratory mechanical testing. The goals of reproducibility and accessibility were achieved by selecting PLA, since it is commonly and readily printable using standard 3D printers. To assess the feasibility of PLA-based 3D-printed bone section simulants, specimens were designed to replicate natural bone sections extracted previously from human cadaveric skulls. The bone sections were printed in different geometries and print orientations to reveal the process-property interrelationship. The results showed the promising potential of PLA as a suitable replacement for human bone in laboratory testing while pointing to the effect of printing orientation and sample geometry on flexural properties. The latter was highly dependent on the sample geometry that gave rise to changes in the section properties and minimally reliant on the printing orientation due to the infill percentage. The difference between the flexural properties of the replicas investigated herein and their natural counterparts (previously reported elsewhere) is attributed to the homogenization approach employed in this study, neglecting the sandwich structure of natural skulls. It is suggested that future research seeks to replicate the exact section properties while using other additive manufacturing techniques. Additionally, PLA was used to additively manufacture full-scale human skulls while exploring different sectioning strategy. The printed and assembled skulls were submitted to lateral compression and frontal penetration tests, and the results were compared to whole 3D-printed skulls. The sectioned skulls reveal the effect of sutures on the structural and mechanical behavior of the 3D-printed skulls while providing a feasible additive manufacturing strategy. The outcomes of this research pave the road for future research emphasizing additive manufacturing of polymeric materials and the development of complete head surrogates for laboratory testing at a broad range of loading conditions.

## Figures and Tables

**Figure 1 polymers-15-00058-f001:**
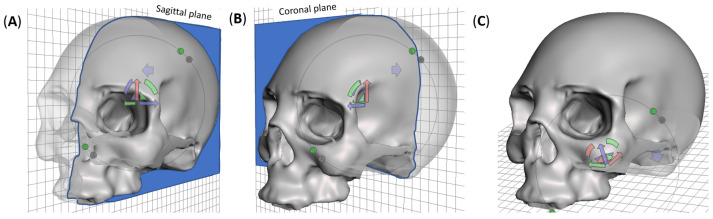
Digital sectioning strategies for additive manufacturing and mechanical characterization of the skull surrogate, showing (**A**) sagittal cut, which yields symmetrical left-right hemi-skulls, (**B**) coronal cut, which splits the skull in frontal and occipital hemi-skulls, and (**C**) whole skull, presenting an opening in the base of the occipital region.

**Figure 2 polymers-15-00058-f002:**
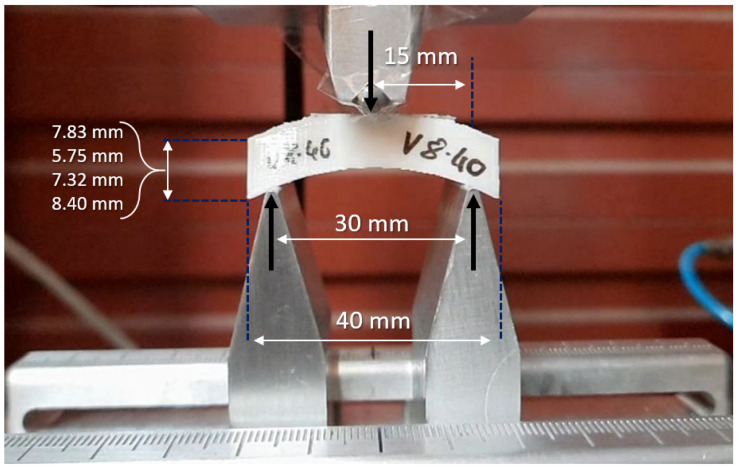
Experimental setup for 3-point flexural tests with Instron Universal Mechanical Testing Machine 3366. The nominal span and thickness of specimens are indicated in the figure, while the nominal depth was 10 mm.

**Figure 3 polymers-15-00058-f003:**
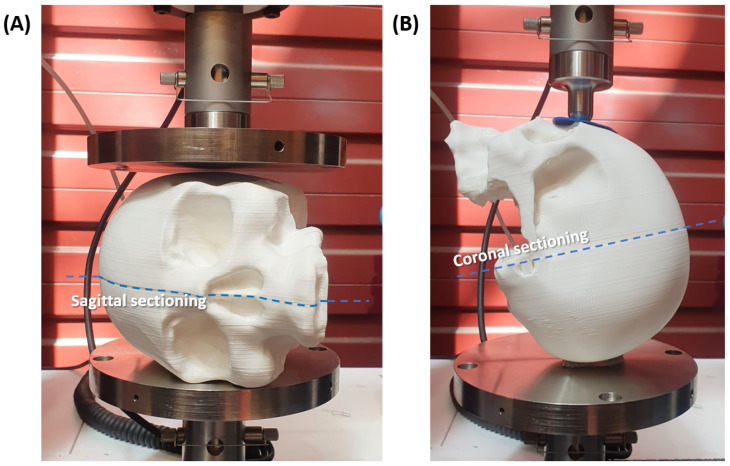
Experimental setup for (**A**) lateral compressive test, (**B**) frontal point-loading test. Anti-slip rubber pieces positioned on the contacting points of the skull with the testing fixture.

**Figure 4 polymers-15-00058-f004:**
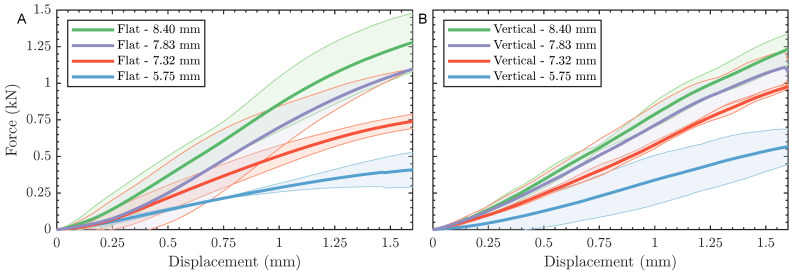
Force-displacement curves showing 3-point bending test results for (**A**) specimens printed on their flat side, and (**B**) specimens printed on their curved side. Each curve shows a solid line for the average of three measurements surrounded with a shaded area corresponding to ±2 SDM.

**Figure 5 polymers-15-00058-f005:**
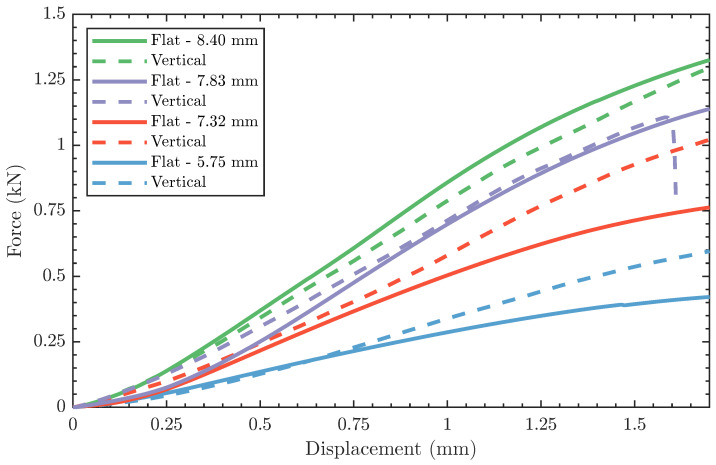
Average force-displacement curves of flat-printed and vertically-printed specimens.

**Figure 6 polymers-15-00058-f006:**
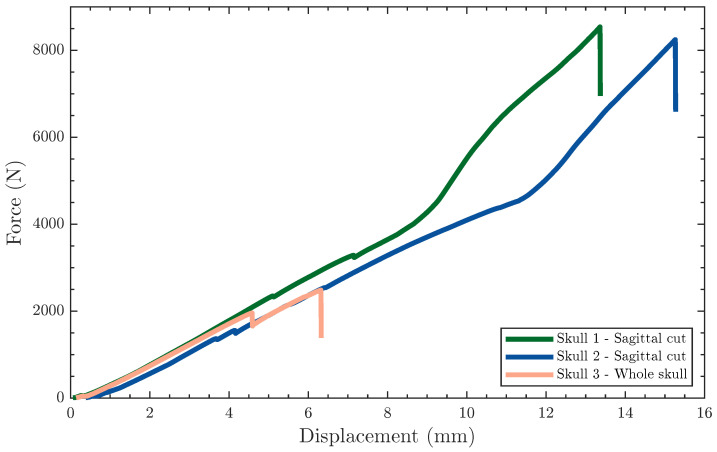
Force-displacement curves showing the response of skull surrogates to lateral compressive tests.

**Figure 7 polymers-15-00058-f007:**
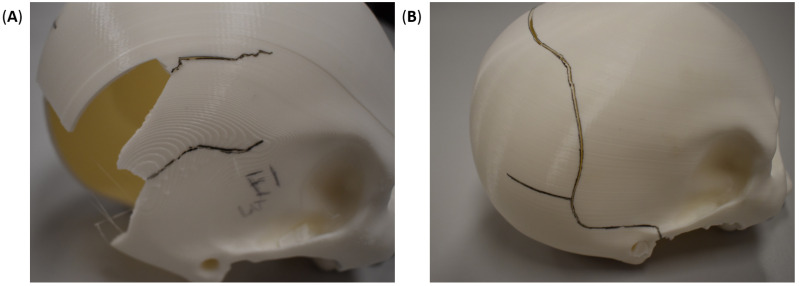
Fracture patterns (shown in black lines) in 3D-printed (**A**) sagittal-cut and (**B**) whole skulls after lateral compressive testing.

**Figure 8 polymers-15-00058-f008:**
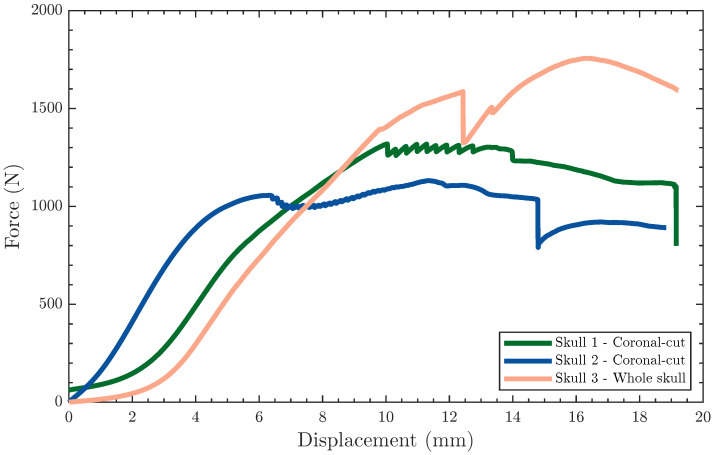
Force-displacement curves showing the response of the surrogate skulls to frontal point-loading tests.

**Figure 9 polymers-15-00058-f009:**
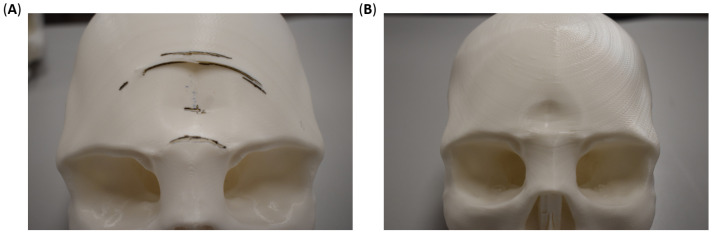
Fracture patterns (shown in black lines) in 3D-printed (**A**) coronal-cut and (**B**) whole skulls after lateral compressive testing.

**Table 1 polymers-15-00058-t001:** Reference skull bone dimensions used in setting the depth and thickness of the 3D printed surrogate bone sections, obtained from [49].

	Skull 1 Bare	Skull 1 Covered	Skull 2 Bare	Skull 2 Covered
Thickness (mm)	7.32 ± 1.87	5.75 ± 1.84	7.83 ± 1.63	8.40 ± 1.96
Depth (mm)	10.12 ± 0.21	10.08 ± 0.32	10.46 ± 0.23	10.09 ± 0.32

**Table 2 polymers-15-00058-t002:** Flexural properties of 3D printed samples compared to published results of natural bone.

	8.40 mm		7.83 mm		7.32 mm		5.75 mm	
	Flat	Vertical	Flat	Vertical	Flat	Vertical	Flat	Vertical
Bending Modulus	2.71 GPa	2.47 GPa	2.68 GPa	2.80 GPa	2.10 GPa	2.32 GPa	1.81 GPa	3.35 GPa
Ref. Bending Modulus ^1^	3.95 ± 0.89 GPa		2.74 ± 1.3 GPa		1.70 ± 0.71 GPa		2.28 ± 0.81 GPa	
Relative Difference ^2^	46.1%	60.3%	2.1%	2.2%	19.1%	26.8%	26.1%	31.9%
Bending Strength	123 MPa	119 MPa	111 MPa	111 MPa	92 MPa	113 MPa	102 MPa	133 MPa
Ref. Bending Strength ^1^	99 ± 14 MPa		53 ± 13 MPa		42 ± 14 MPa		68 ± 13 MPa	
Relative Difference ^2^	19.8%	16.5%	52.2%	52.2%	54.2%	62.7%	33.3%	49.0%

^1^ As reported by Lee et al. [49]. ^2^ Calculated based on the differences in PLA-based samples from this work and scalp bone samples results from work cited.

**Table 3 polymers-15-00058-t003:** Mechanical properties of specimens subjected to lateral compressive tests.

	Skull 1—Sagittal Cut	Skull 2—Sagittal Cut	Skull 3—Whole Skull
Stiffness (N/mm)	502.1	484.6	486.7
Displacement at fracture (mm)	13.32	15.26	6.321
Max. Force (N)	8539	8246	2468

**Table 4 polymers-15-00058-t004:** Mechanical properties of specimens subjected to frontal point-loading tests.

	Skull 1—Coronal Cut	Skull 2—Coronal Cut	Skull 3—Whole Skull
Stiffness (N/mm)	231.1	278.3	235.8
Displacement at yielding (mm)	10.02	6.317	12.37
Max. Force (N)	1319	1131	1756

## Data Availability

The data presented in this study are available upon reasonable request from the authors.

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
