# Peer review of "Additive Manufacturing and Mechanical Characterization of PLA-Based Skull Surrogates"

_polymers, 2022, doi:10.3390/polym15010058_

Round 1
Reviewer 1 Report
1- The perspective of this work should be written carefully.
2- The other mechanical properties are useful for example thermal conductivity, hardness vigers, surface roughness, compressive strength?3- How the author measure he thicknesses of the layers?
4- Why the author didn’t use Four-point Bending tests?
5- Please, refer to these refs for mechanical properties.
DOI: https://doi.org/10.1088/1742-6596/1795/1/012059
DOI: https://doi.org/10.1088/1742-6596/1795/1/012052
Reviewer 2 Report
Journal: Polymers (ISSN 2073-4360)
Manuscript ID: polymers- 2105080
Type: Article
Title: Additive Manufacturing and Mechanical Characterization of PLA-Based Skull Surrogates.
Authors: Ramiro Mantecón * , Miguel Marco , Ana Muñoz Sanchez , George Youssef , José Díaz Álvarez , Henar Miguélez.
a) Introduction: In the literature, add more than three to the number of authors who have worked on their subject and achieved results.
b) Write the objective of the present work clearly.
c) Why the authors didn’t measure the thermal conductivity of the sampls?.
d) The other mechanical properties are useful for example thermal conductivity, hardness vigers, surface roughness, compressive strength?
e) For references, choose recent refs. Please, refer to these refs for mechanical properties.
DOI: https://doi.org/10.1088/1742-6596/1795/1/012059
DOI: https://doi.org/10.1016/j.jallcom.2017.02.117
Best Regards
Reviewer 3 Report
The study is about additive manufacturing and mechanical characterization of PLA-based skull surrogates. The manuscript is very well written.
The results are technically correct and backed up by plenty of adequate discussions from literature.
If it is possible, adding a schematic presentation of method or a graphical abstract can give high impact to the manuscript.
The superiority of the numerical results of the obtained mechanical properties to the literature results can be added to the discussion section.
I believe that the inferences will make an important contribution to science and literature.
Reviewer 4 Report
The authors worked on a manuscript entitled "Additive Manufacturing and Mechanical Characterization of PLA-Based Skull Surrogates", which is novel and benefits the AM community. The following points need to be addressed before publication.
1) Line 20 - "Brain injuries are commonly associated with head trauma due to blunt... Provide appropriate reference
2) The introduction must be 02 pages
3) Lines 118-139 are a general discussion about FDM. Restrict only 10-12 lines.
4) Lines 141-142 provide appropriate reference
5) Section 2.2 add more recent papers related to PLA.
6) Line 214 Why do authors apply 10 kN load cell?
7) Line 218 The 3-point bending tests were performed at 10 mm/min. Any reason for a particular value?
8) What about the surface finish of the final human skulls made by PLA?
9) What is layer thickness during printing?
10) Is there a delamination issue that occurred in the printing?
Round 2
Reviewer 4 Report
Dear Authors,
Now the manuscript is suitable for publication.